# Accuracy and Efficiency of Right-Lobe Graft Weight Estimation Using Deep-Learning-Assisted CT Volumetry for Living-Donor Liver Transplantation

**DOI:** 10.3390/diagnostics12030590

**Published:** 2022-02-25

**Authors:** Rohee Park, Seungsoo Lee, Yusub Sung, Jeeseok Yoon, Heung-Il Suk, Hyoungjung Kim, Sanghyun Choi

**Affiliations:** 1Department of Radiology, Research Institute of Radiology, Asan Medical Center, University of Ulsan College of Medicine, Seoul 05505, Korea; bbakhyung91@gmail.com (R.P.); hjkim.radiology@gmail.com (H.K.); edwardchoi83@gmail.com (S.C.); 2Department of Convergence Medicine, Asan Medical Center, University of Ulsan College of Medicine, Seoul 05505, Korea; asmilez.sung@gmail.com; 3Department of Brain and Cognitive Engineering, Korea University, Seoul 08308, Korea; wltjr1007@korea.ac.kr (J.Y.); hisuk@korea.ac.kr (H.-I.S.); 4Department of Artificial Intelligence, Korea University, Seoul 08308, Korea

**Keywords:** deep learning, CT volumetry, segmentation, living right liver donors

## Abstract

CT volumetry (CTV) has been widely used for pre-operative graft weight (GW) estimation in living-donor liver transplantation (LDLT), and the use of a deep-learning algorithm (DLA) may further improve its efficiency. However, its accuracy has not been well determined. To evaluate the efficiency and accuracy of DLA-assisted CTV in GW estimation, we performed a retrospective study including 581 consecutive LDLT donors who donated a right-lobe graft. Right-lobe graft volume (GV) was measured on CT using the software implemented with the DLA for automated liver segmentation. In the development group (*n* = 207), a volume-to-weight conversion formula was constructed by linear regression analysis between the CTV-measured GV and the intraoperative GW. In the validation group (*n* = 374), the agreement between the estimated and measured GWs was assessed using the Bland–Altman 95% limit-of-agreement (LOA). The mean process time for GV measurement was 1.8 ± 0.6 min (range, 1.3–8.0 min). In the validation group, the GW was estimated using the volume-to-weight conversion formula (estimated GW [g] = 206.3 + 0.653 × CTV-measured GV [mL]), and the Bland–Altman 95% LOA between the estimated and measured GWs was −1.7% ± 17.1%. The DLA-assisted CT volumetry allows for time-efficient and accurate estimation of GW in LDLT.

## 1. Introduction

Living-donor liver transplantation (LDLT) is an effective therapeutic option for patients with end-stage liver disease [1]. An adequate graft mass is a major component of a successful LDLT. The use of small-for-size grafts, with graft-to-recipient weight ratios of less than 0.8 to 1%, is associated with graft malfunction, while an insufficient remnant liver mass after harvesting a graft may threaten donor safety [2,3]. Therefore, the accurate preoperative estimation of graft weight is a prerequisite step in LDLT to ensure the safety of both recipients and donors. 

CT volumetry has been widely used for preoperative graft volume measurement in LDLT [4,5,6,7,8,9,10,11,12]. Graft weight is usually estimated using CT-measured graft volume and a volume-to-weight conversion formula [10,11,12,13,14,15]. Although several studies have assessed the performance of CT volumetry in graft weight estimation [5,7,8,10,11,12,13,14,15,16,17], these studies had limitations. The volume-to-weight conversion formulae used in the previous studies were not reliable since they were derived from small study populations (i.e., ≤16 subjects) [10,11,14], pathologic liver conditions [10], or the assumption that the liver and water have the same density [12,13,15], which may have led to biased estimations of graft weight. Moreover, previous studies assessed the correlations or mean differences between the estimated and actual graft weights, but did not evaluate the measurement error of CT volumetric graft weight estimation [5,6,7,8,9,10,11,12], which is important to predict the range of actual graft weight in individual LDLT donors. 

One obstacle that limits the clinical use of CT volumetry has been the time-consuming organ segmentation process. Recently, deep learning has emerged as a method for automated image analysis. Recent studies have demonstrated that a deep-learning algorithm (DLA) enables fully automated segmentation of the liver using CT images with high accuracy, allowing for automated liver volume measurement without user interaction [18]. Thus, the application of DLA for CT-based liver segmentation would dramatically improve the time efficiency of CT volumetry in estimating graft weight for LDLT. 

Therefore, the purpose of our study was to construct a graft volume-to-weight conversion formula, and to evaluate the accuracy and time efficiency of DLA-assisted CT volumetry for estimating graft weight in a large cohort of living liver donors who donated right-lobe liver grafts.

## 2. Material and Method

This retrospective study was approved by our institutional review board, which waived the requirement for patients’ informed consent.

### 2.1. Study Population 

We retrospectively and consecutively enrolled living liver donors in our institution who donated right-lobe grafts from 2013 to 2015. Eligible donors were those who had undergone CT examinations within 3 months before liver donation and had data on intraoperative graft weight measurement. A total of 581 donors who satisfied the eligibility criteria comprised the study population (Figure 1). The study population was then divided into the development (liver donation in 2013, *n* = 207) and validation (liver donation from 2014 to 2015, *n* = 374) groups. A subset of 50 donors who were randomly selected from the validation group comprised the subgroup used to assess inter-reader agreement in graft volume measurement. 

### 2.2. CT Examination

CT examinations were performed using various CT scanners and techniques (Appendix A). CT scans were obtained using 16-channel (Sensation 16, Simens Healthineers, Erlangen, Germany), 64-channel (Definition AS, Siemens Healthineers or Lightspeed VCT, GE Healthcare, Milwaukee, WI, USA), or 128-channel (Definition Flash, Siemens Healthineers) scanners. Portal venous phase images were obtained 76 s after intravenous contrast administration with tube voltages of 100 or 120 kVp, tube currents of 200–440 mA with automatic exposure control, matrix of 512 × 512, and section thicknesses of 3 or 5 mm with no gap. The total number of images ranged from 95 to 115 for the examinations with 5 mm slice thickness, and ranged from 155 to 188 for those with 3 mm slice thickness.

### 2.3. Graft Volume Measurement Using a Deep Learning Algorithm

The graft volume was measured by a reader who was a third-year radiology resident (P.R.) on portal venous phase CT images. The reader first analyzed CT data from the first 30 donors in the development group, together with an experienced radiologist (L.S.S, with 23 years of experience in abdominal imaging) for training purposes. The CT data were analyzed using the GoCDSS software (SmartCareworks Inc., Seoul, Korea), which applied a DLA for automated liver segmentation. The detailed information of the DLA was described previously [18], and its source code is provided at https://github.com/seungsoolee0007/liver_spleen_segmentation (accessed on 22 February 2022). Briefly, the algorithm performed whole-liver segmentation, excluding large hepatic vessels, with a dice similarity score of 97% in a computation time of 33 s for a typical abdominal CT examination [18]. Once the CT data were uploaded, the software automatically performed liver segmentation. Then, the reader reviewed CT images along with the deep-learning-generated liver segmentation results and corrected any segmentation errors. The liver volumes measured by automated segmentation by the deep learning algorithm and those measured after the reader’s correction were recorded. The reader defined the resection plane for the right-lobe graft based on the Cantlie line by drawing two dividing lines (one along the main axis of the middle hepatic vein, superiorly, and the other along the imaginary line between the gallbladder and inferior vena cava, inferiorly) on the selected images (Figure 2). The software completed the resection plane by interpolation of the two dividing lines. The volumes of the whole liver and right-lobe liver graft were automatically calculated by summation of the area multiplied by the slice interval. The times required for reviewing CT images, correcting segmentation errors, and defining the resection plane were recorded. To assess inter-reader agreement, the second reader (L.S.S) independently measured graft volume in a subset of 50 donors from the validation group. 

### 2.4. Clinical and Pathologic Data and Intraoperative Graft Weight Measurement

Clinical data including age, sex, height, weight, and body mass index (BMI) were obtained on the day of the CT examinations. The degree of hepatic steatosis (HS) was assessed by pathologic analysis of ultrasonography-guided percutaneous liver biopsy specimens, which was performed 1–78 days (median, 17 days) before liver donation as a part of donor workup. The degree of HS was graded as none (<5%), mild (5–33%), moderate (34–66%), or severe (>66%), as defined by the Non-alcoholic Steatohepatitis Clinical Research Network scoring system [19]. The graft weight was measured during the donor hepatectomy and served as the reference standard in our study. The donor hepatectomy was performed as described previously [5]. Briefly, the demarcation line of the right lobe was drawn based on the color change of the liver surface that occurred during temporary clamping of the right portal vein and right hepatic artery. Parenchymal dissection was performed with the middle hepatic vein as the anatomic landmark, i.e., along the right (graft without middle hepatic vein) or left (graft with middle hepatic vein) side of the middle hepatic vein. Dissection of the dorsal part of the liver was performed using a hanging maneuver that allowed transection of the liver parenchyma down to the inferior vena cava. After harvesting a right-lobe graft, the surgeons shook the excised graft to spill out the remaining blood, waited a few seconds for natural drainage, then measured the blood-free graft weight on an electronic laboratory scale (FD 110; Excel Precision, New Taipei City, Taiwan). 

### 2.5. Statistical Analysis 

The characteristics of the development and validation groups were compared using independent t-tests or Fisher’s exact tests. Agreements between the whole-liver volumes automatically measured by DLA and those measured after the radiologist’s correction were evaluated using the Bland–Altman 95% limit of agreement (LOA). Bland–Altman 95% LOAs were expressed as percentages of the measured values and as the mean differences ± 1.96 × standard deviation (SD) of the difference, where the mean difference represented systemic bias, and 1.96 × SD of the difference represented the measurement error. In the developmental group, to evaluate the confounding effect of HS on the graft weight, multivariable linear regression analysis was performed by including the HS- and CT-measured graft volume as independent variables and graft weight as the dependent variable. Then, the formula to convert CT-measured graft volume to graft weight was built using univariable linear regression analysis. In the validation group, the graft weight was estimated using CT-measured graft volume, and the conversion formula derived from the development group. The agreement between the estimated and measured graft weights was then assessed using the concordance correlation coefficient (CCC) and Bland–Altman 95% LOA. In the subgroup including 50 donors in the validation group, the inter-reader agreement in the graft volumes between the two radiologists was assessed using the CCC and Bland–Altman 95% LOA. To assess the factors influencing the magnitude of error in graft weight estimation, multivariable linear regression analysis was performed in the validation cohort; this included the age, sex, BMI, HS, interval between CT scan and liver donation, and type of liver graft (right-lobe graft with or without middle hepatic vein) as independent variables; and the percentage difference between the estimated and measured graft weights, i.e., (estimated graft weight—measured graft weight)/measured graft weight, as the dependent variable. Statistical analyses were performed using IBM SPSS Statistics for Windows, version 21.0 (IBM Corp., Armonk, NY, USA) and MedCalc version 14.8.1 (MedCalc, Ostend, Belgium). *p*-values < 0.05 were considered to indicate significant differences.

## 3. Results

### 3.1. Characteristics of the Study Population

Table 1 summarizes the characteristics of the study population. The study population included 581 donors (413 men and 168 women; mean age, 27.7 years; age range, 17–54 years). Most donors had non-steatotic liver, and clinically relevant HS was present in 89 (15.3%) donors, with mild HS in 87 (15.0%) and moderate HS in two (0.3%) donors. The development and validation groups included 207 (132 men and 75 women; mean age, 27.6 years; age range, 18–54 years) and 374 (281 men and 93 women; mean age, 27.8 years; age range, 17–52 years) donors, respectively. The developmental and validation groups showed significant differences in sex (*p* = 0.004), BMI (*p* = 0.015), body weight (*p* = 0.022), and intraoperatively measured graft weight (*p* = 0.004). 

### 3.2. Liver Segmentation and Graft Volume Estimation Using DLA

DLA-generated automated segmentation of the whole liver showed segmentation errors requiring radiologists’ correction in 166 (28.6%) donors. Most segmentation errors were minor and were associated with a short correction time (the mean time required for the radiologists’ correction ± standard deviation [SD], 12.8 ± 33.6 s) and a small change in volume (Bland–Altman 95% LOAs, 0.05% ± 3.0% of the measured liver volume). The mean process time, including the DLA-generated segmentation review, and correction and division of the segmented liver, was 1.8 ± 0.6 min (range, 1.3–8.0 min). 

### 3.3. Construction of Graft Volume-to-Weight Conversion Formula in the Development Group

In the development group, the CT-measured graft volume and intraoperatively measured graft weight ranged from 454.9 mL to 1187.0 mL (mean ± SD, 1249.0 ± 237.9 mL) and from 420.0 g to 1025.0 g (mean ± SD, 728.1 ± 123.5 g), respectively. In multivariable linear regression analysis, HS did not have a significant effect on graft weight (coefficient, −0.34; *p* = 0.667) after accounting for the effect of graft volume on graft weight (coefficient, 0.655; *p* < 0.001). Therefore, we constructed the formula to convert CT-measured graft volume to graft weight in the entire development group, without excluding donors with HS (Figure 3). The conversion formula was as follows: estimated graft weight (g) = 206.3 + 0.653 × CT-measured graft volume (mL) (*r* = 0.878, *p* < 0.001). 

### 3.4. Agreement between the Estimated and Measured Graft Weights in the Validation Group

In the validation group, the CT-measured graft volume, estimated graft weight, and intraoperatively measured graft weight ranged from 722.9 mL to 2259.6 mL (mean ± SD, 1281.9 ± 233.4 mL), from 520.0 g to 1153.7 g (mean ± SD, 743.6 ± 104.0 g), and from 456.0 g to 1400.0 g (mean ± SD, 760.2 ± 131.2 g), respectively. The CCC for the agreement between the estimated and measured graft weights was 0.834 (95% confidence interval [CI], 0.804 to 0.860) (Figure 4). The Bland–Altman 95% LOA was −1.7% ± 17.1% (*p* = 0.002 for the difference of mean bias from zero), indicating a mean bias of −1.7% and measurement error of 17.1% of the graft weight (Figure 5). 

The inter-reader agreement for graft volume measurement was assessed in the subset of 50 donors in the validation cohort. The CCC for the agreement in the graft volume measurement between the two readers was 0.998 (95% CI, 0.996–0.999), and the Bland–Altman 95% LOA was 0.2% ± 1.8% (*p* = 0.069). 

### 3.5. Factors Associated with Differences in Estimated and Measured Graft Weights 

In the validation cohort, multivariable linear regression analysis revealed that sex (coefficient, −1.73; *p* = 0.001) and BMI (coefficient, −0.2; *p* < 0.001) showed significant independent associations with the percentage difference between the estimated and measured graft weights, while age (*p* = 0.076), HS degree (*p* = 0.577), interval between CT and liver donation (*p* = 0.111), and type of liver graft (*p* = 0.279) did not. The Bland–Altman 95% LOAs between the estimated and measured graft weights were −2.6% ± 16.8% (*p* < 0.001) and 0.9% ± 16.8% (*p* = 0.306) for men and women, respectively. When donors were sub-grouped according to BMI, the Bland–Altman 95% LOAs between the estimated and measured graft weights were −1.3% ± 16.2% (*p* = 0.009) and −4.2% ± 17.0% (*p* <.001) for donors with BMI <25 kg/m^2^ and overweight or obese donors with BMI ≥25 kg/m^2^, respectively. 

## 4. Discussion

Our study evaluated the efficiency and accuracy of CT volumetry using a DLA for the preoperative estimation of right-lobe graft weight in LDLT. We found that the DLA allowed for a time-efficient measurement of graft volume on CT. The CT data analysis with the DLA was performed as a background process so that the reader reviewed CT images with DLA-generated liver segmentation results. The DLA enabled highly accurate segmentation of the liver. The DLA-generated liver segmentation results did not require additional correction in approximately 70% of donors; moreover, minor segmentation errors, which were rapidly corrected by the reviewing radiologist, were observed in only 30% of donors. As a result, CT volumetric assessment of right-lobe grafts was rapidly performed in an average process time of 1.8 min. 

We developed the graft volume-to-weight conversion formula in a large number of donors (i.e., 207 donors) in the development group. In the validation group, graft weights that were estimated using the CT-measured graft volume and conversion formula showed an overall good agreement with the measured graft weights (CCC = 0.834). The Bland–Altman 95% LOA indicated a measurement error of graft weight of ±17.1%. Although this measurement error appears large, the range encompasses most (i.e., 95%) of the actual difference between the estimated and measured graft weights. Compared to our study, previous studies reported even greater differences between estimated and measured graft weights, ranging from −48.2% to 66.2% [5,8,12]. As suggested by previous studies, multiple factors may have contributed to the error in CT volumetric estimation of graft weight, including mismatches between expected and actual resection planes [6,12], graft dehydration [7,12], and variable amounts of blood remaining in the graft [8]. 

Our validation results showed a small but significant bias in the estimated graft weights, indicating that graft weight was underestimated by 1.7% in the validation group. The mean bias of 1.7% in our study was smaller than those reported previously (i.e., −9.8% to 2.4% of graft weight) [5,8], which may have been partly due to our use of a conversion formula developed in a larger study population. Though not yet fully understood, the bias in the estimated graft weights in the validation group may have been related to different characteristics between the development and validation groups. Our study showed that sex and BMI were significantly associated with the percentage difference between the estimated and measured graft weights. Graft weights tended to be underestimated in men (mean bias, −2.6%) and in donors with a higher BMI (mean bias, −4.2%). Thus, a significantly higher proportion of men and donors with a higher BMI in the validation group than in the development group may have led to a small underestimation of graft weight in the validation group. 

We observed nearly perfect inter-reader agreement in graft volume measurement between the two readers (CCC = 0.998; Bland–Altman 95% LOA = −1.6% to 2.0% of the measured volume). This finding is noteworthy given the different level of experience between the two readers (i.e., third-year radiology resident vs. abdominal radiologist with 23 years of experience). The use of automated liver segmentation with the DLA may help to reduce inter-reader variability in liver segmentation, making CT-based graft volume measurement simple enough for a less experienced reader to learn after a short training session. 

In our study, HS did not show a significant confounding effect on the association between graft volume and graft weight in the development cohort. In addition, the degree of HS was not a significant factor for the percentage difference between the estimated and measured graft weights. However, this finding should be interpreted with caution as most donors in our study population had no or mild HS. Despite some controversies, there have been a few prior reports suggesting the association of HS with increased liver volume [20,21,22]. Therefore, our results may not be directly generalizable to donors with moderate-to-severe HS.

Our study had several limitations. First, the retrospective design may be subject to selection bias and bias from missing data, despite our efforts to minimize such biases by enrolling consecutive donors. Second, we evaluated only right-lobe grafts since that is the preferred graft for LDLT to meet recipients’ metabolic demands [23,24]. Thus, the measurement error range for CT volumetric graft weight estimation in our study may not be directly applicable to other types of liver graft. Finally, the development and validation groups in our study were enrolled in the same institution. External validation in a completely different population may have provided more conclusive validation results.

## 5. Conclusions

In conclusion, we proposed a graft volume-to-weight conversion formula for pre-operative CT volumetric estimation of graft weight in LDLT. The DLA-assisted CT volumetry provided a time-efficient and accurate estimation of graft weight in LDLT. The measurement error of the CT volumetric estimation of right-lobe graft weight was approximately 17% of the graft weight. 

## Figures and Tables

**Figure 1 diagnostics-12-00590-f001:**
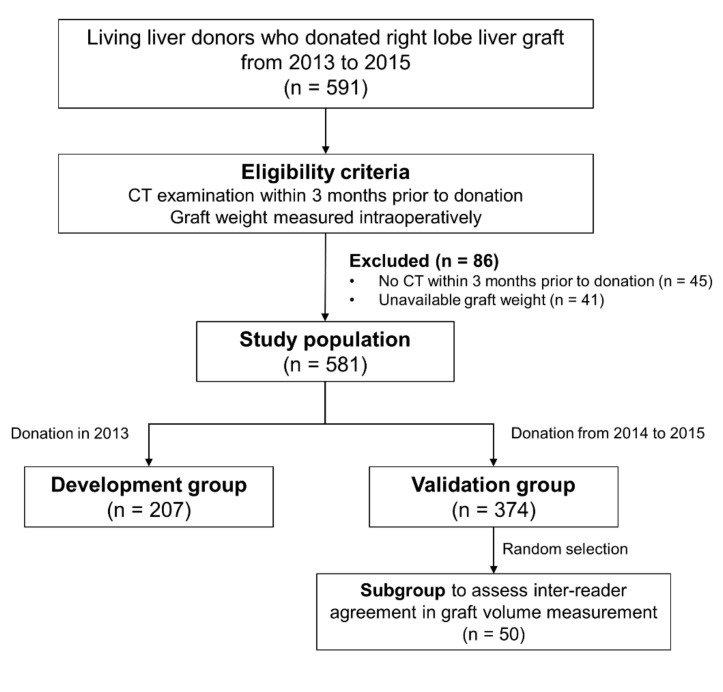
Flow diagram of the study population.

**Figure 2 diagnostics-12-00590-f002:**
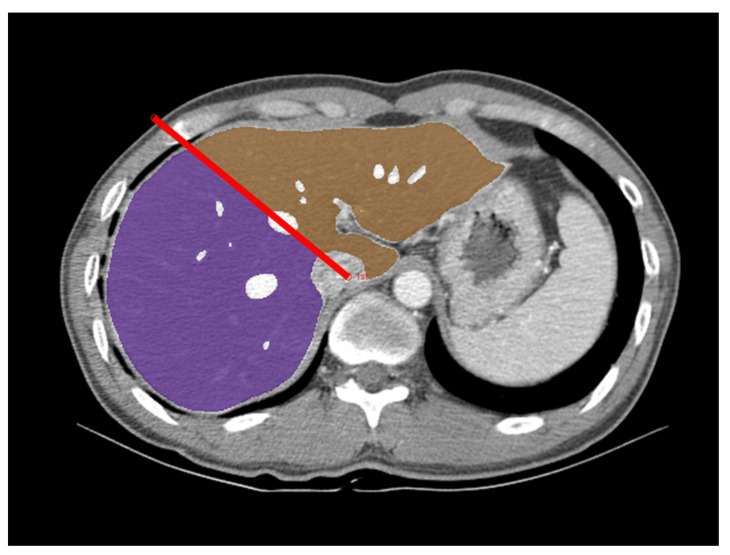
Measurement of the right-lobe graft volume using deep-learning-algorithm-assisted CT volumetry. An axial portal venous phase CT image in a 44-year-old male donor is overlaid with a right-lobe mask (purple), a left lobe mask (brown), and a dividing line (red line). CT image data were first processed by the deep learning algorithm for whole liver segmentation. The radiologist reviewed these results, corrected any segmentation errors, and defined the resection plane for the right-lobe graft by drawing the dividing lines.

**Figure 3 diagnostics-12-00590-f003:**
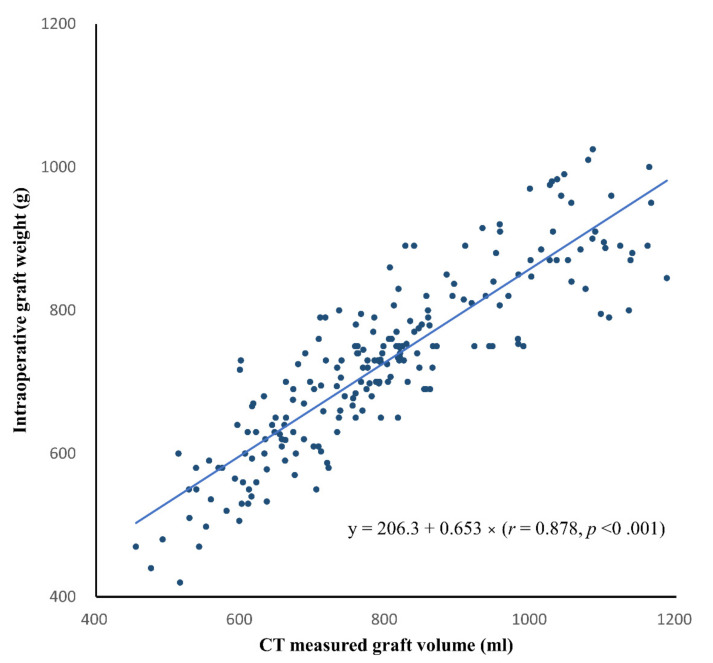
Scatter plot of the CT-measured graft volume versus intraoperative graft weight in the development group. The solid line indicates the best-fit regression line. The linear regression equation is also shown.

**Figure 4 diagnostics-12-00590-f004:**
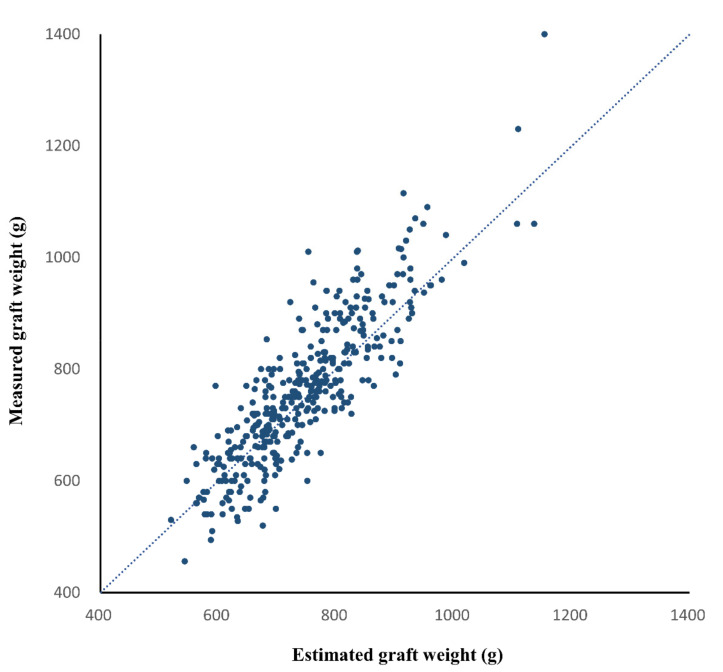
Scatter plot of the estimated and measured graft weights in the validation group. The dashed line is the reference line indicating complete agreement. The concordance correlation coefficient between the estimated and measured graft weights was 0.834 (95% confidence interval, 0.804 to 0.860, *p* < 0.001).

**Figure 5 diagnostics-12-00590-f005:**
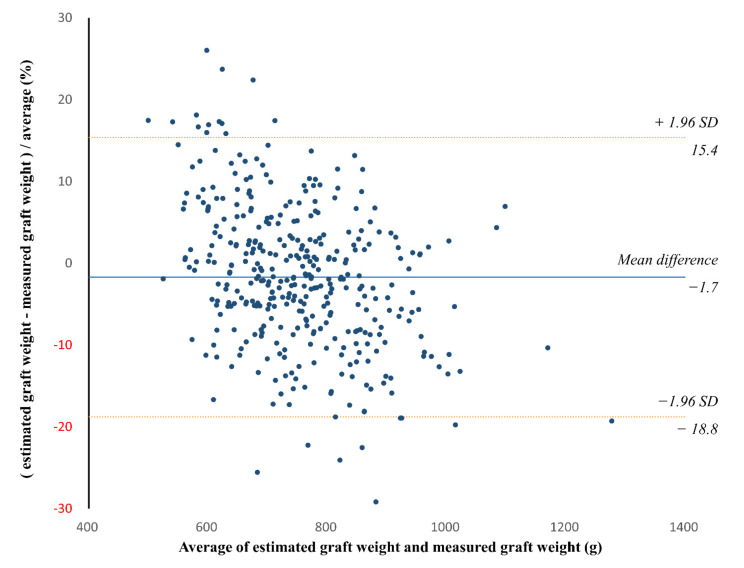
Bland–Altman plot of the agreement between the estimated and measured graft weights in the validation group. The solid line indicates the mean difference, while the dashed lines indicate the upper and lower limits of the 95% limits of agreement. The Bland–Altman 95% limit of agreement (LOA) was −1.7% ± 17.1% (*p* = 0.002 for the difference of mean bias from zero). SD = standard deviation.

**Table 1 diagnostics-12-00590-t001:** Characteristics of the study population.

KERRYPNX	Total	Developmental Group	Validation Group	*p*-Value *
Number of patients	581	207	374	
Age (years) ^†^	27.7 ± 7.2	27.6 ± 6.9	27.8 ± 7.3	0.661
Sex				
Male	413 (71.1)	132 (63.8)	282 (74.8)	0.004
Hepatic steatosis				0.816
None	492 (84.7)	177 (85.5)	315 (84.2)	
Mild	87 (15.0)	29 (14.0)	58 (15.5)	
Moderate	2 (0.3)	1 (0.5)	1 (0.3)	
BMI (kg/m^2^) ^†^	22.9 ± 2.9	22.5 ± 2.9	23.1 ± 2.9	0.015
Height (cm) ^†^	170.4 ± 8.1	169.9 ± 8.4	170.7 ± 7.9	0.268
Weight (kg) ^†^	66.8 ± 11.4	65.4 ± 11.7	67.6 ± 11.1	0.022
Type of liver graft				0.151
RLG without MHV	559 (96.2)	196 (94.7)	363 (97.1)	
RLG with MHV	22 (3.8)	11 (5.3)	11 (2.9)	
Interval between CT and graft donation (days) ^†^	30.6 ± 19.0	30.9 ± 18.8	30.4 ± 19.1	0.728
Graft weight (g) ^†^	748.8 ± 129.3	728.1 ± 123.5	760.2 ± 131.2	0.004

Unless otherwise indicated, data are shown as the number of patients, with percentages in parentheses. RLG = right-lobe graft, MHV = middle hepatic vein. * *p*-values for comparisons between the development and validation groups; ^†^ Mean ± standard deviation.

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
