# Peer review of "Accuracy and Efficiency of Right-Lobe Graft Weight Estimation Using Deep-Learning-Assisted CT Volumetry for Living-Donor Liver Transplantation"

_diagnostics, 2022, doi:10.3390/diagnostics12030590_

Round 1
Reviewer 1 Report
Authors, through a retrospective study, aims at deriving a graft volume-to-weight conversion formula and to evaluate the accuracy and time efficiency of deep learning segmentation -assisted CT volumetry for estimating graft weight in a large cohort of living liver donors who donated right-lobe liver grafts.
The chosen DL approach is the GOCDSS by SmartCareWorks.
The choise of using a specific commercial deepl learning based liver segmentation tool, limits too much the conclusions in assessing the efficacy of using DL to estimate the liver weight.
Some minor changes are needed:
- Sec 2.2: add the total number of images for each patient and slice thickness
- sec 3.3 the weight range is from 767.8 ml to 1880.7 ml that doesn't correspond to the values reported in Fig. 3
Author Response
Point by point responses to reviewers’ comments
Reviewer #1
R1-1 Sec 2.2: add the total number of images for each patient and slice thickness
- We added total number of images according to slice thickness.
R1-2 sec 3.3 the weight range is from 767.8 ml to 1880.7 ml that doesn't correspond to the values reported in Fig. 3
- We appreciate the reviewer for indicating our mistake. We have corrected the text.
Reviewer 2 Report
The paper entitled “Accuracy and efficiency of right-lobe graft weight estimation using deep learning-assisted CT volumetry for living donor liver transplantation” by. Rohee Park et al. evaluates the efficiency and accuracy of deep learning algorithm and CT volumetry in pre-operative graft weight.
In section 2.1. Study population it is mentioned that the validation group is 374 patients form 581, however, it is over 60% from the target group, it is unclear why the authors have chosen so a large group for validation.
In the section „2.2. CT examination” is shown the acquisition stage, the authors should add the quality and resolution of the images.
In the section „2.3. Graft volume measurement using a deep learning algorithm” is added the processed image with the proposed algorithm, please add a pseudocode or logic scheme of it.
In the section „2.5. Statistical analysis” the Bland–Altman is a statistical method for analyzing the agreements between the whole liver volume automatically measured by DLA and those measured after the radiologist’s correction. The paper should be described the method of how was computed the volume.
The quality of Figures 3, 4, and 5 should be improved.
The results are compared with studies from 2010 (reference 8) 2003 (reference 12) however these are old taking into account that the study is based on a deep learning algorithm.
Please create a separate conclusion section.
Author Response
Point by point responses to reviewers’ comments
Reviewer #2
R2-1 In section 2.1. Study population it is mentioned that the validation group is 374 patients form 581, however, it is over 60% from the target group, it is unclear why the authors have chosen so a large group for validation.
- It is a usual approach to have a larger development group than the validation group. However, we decided to have a larger validation group than the development in our study because of the following reasons: first, we considered that the 207 donors would be sufficient for constructing volume to weight conversion equation. Second, the main purpose of our study was to evaluate the measurement error in graft weight estimation using CT volumetry, which, we considered, need to be evaluated in a large validation group for a precise estimation.
R2-2 In the section „2.2. CT examination” is shown the acquisition stage, the authors should add the quality and resolution of the images
- According to the reviewer’s suggestion, we provided the resolution (matrix of 512 x 512) of CT images. However, we do not understand what the reviewer exactly means by the quality of CT examination. As we already mentioned in section 2.2 and supplementary table 1, CT examinations were conducted using up-to-date scanner system and standard technique and thus considered to be performed with clinically acceptable image quality.
R2-3 In the section „2.3. Graft volume measurement using a deep learning algorithm” is added the processed image with the proposed algorithm, please add a pseudocode or logic scheme of it.
- We provided source code for the deep learning algorithm
R2-4 In the section „2.5. Statistical analysis” the Bland–Altman is a statistical method for analyzing the agreements between the whole liver volume automatically measured by DLA and those measured after the radiologist’s correction. The paper should be described the method of how was computed the volume.
- We already stated that liver volume was calculated by summation of the area multiplied by the slice interval. According to the reviewer’s recommendation, we additionally stated that the liver volumes measured by automated segmentation by the deep learning algorithm and those measured after the reader’s correction were recorded.
R2-5 The quality of Figures 3, 4, and 5 should be improved.
- According to the reviewer’s suggestion, we have improved the quality of figures
R2-6 The results are compared with studies from 2010 (reference 8) 2003 (reference 12) however these are old taking into account that the study is based on a deep learning algorithm.
- We appreciate the reviewer for the constructive criticism on our study. We tried to find up-to-date references to be compared with our results. However, as we already stated in our manuscript, there was no recent publication on this topic. We speculate that this reflects a lack of advance in CT volumetry until the emergence of deep learning-based algorithm for automated organ segmentation, which highlights the importance of our findings.
R2-7. Please create a separate conclusion section.
- We created a separate conclusion section according to the reviewer’s suggestion.